# Chicory Taproot Production: Effects of Biostimulants under Partial or Full Controlled Environmental Conditions

Gabriele Paglialunga [1,2], Simona Proietti [1], Mariateresa Cardarelli [2], Stefano Moscatello [1], Giuseppe Colla [2,*] and Alberto Battistelli [1,*]

1   Istituto di Ricerca Sugli Ecosistemi Terrestri, Consiglio Nazionale delle Ricerche, 05010 Porano, Italy
2   Department of Agriculture and Forest Sciences, University of Tuscia, 01100 Viterbo, Italy
*   Correspondence: giucolla@unitus.it (G.C.); alberto.battistelli@cnr.it (A.B.); Tel.: +39-(0)761-357536 (G.C.); +39-(0)763-374910 (A.B.)

**Abstract:** Two trials were conducted on chicory (*Cichorium intybus* L.) grown under greenhouse and growth chamber conditions with the aim to evaluate the potential of three biostimulants (seaweed extract (SWE), animal-derived protein hydrolysate (APH), and vegetal-derived protein hydrolysate (VPH) on improving quali-quantitative traits of taproot, in short and out-of-season production cycles. In the greenhouse trial, VPH biostimulant promoted the inulin yield on a per-hectare basis with respect to the untreated control and APH. Taproot fresh weights, dry weights, and diameter in VPH-treated plants increased in comparison with APH-treated ones. SWE-treated plants showed intermediate values of the root production parameters and the inulin yield, with no statistical difference with VPH, APH, and control. In the growth room trial, SWE, VPH, and control showed no significant differences in growth, root yield, and quality. The results demonstrated that VPH can be useful for improving root production and inulin yield of chicory under partial controlled conditions such as in a greenhouse, whereas no benefits of biostimulant applications on crop yield and quality traits were recorded in growth chambers under full control of micro-climate conditions.

**Keywords:** chicory taproots; inulin; plants biostimulants; controlled environment

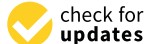



## 1. Introduction

Chicory (*Cichorium intybus* L.) is a cultivated species belonging to the *Asteraceae* family, naturally diffused, and cultivated in many regions of the world. The extensive breeding undergone in centuries has produced several varieties; leaves, stems, and roots of chicory are consumed fresh or cocked, utilized as herbal medicine preparations, or destined for industrial processing [1–3]. Chicory roots are traditionally consumed as vegetable in some regions of Italy and France. Moreover, chicory roots are produced for the extraction of inulin-type fructans, which are well-known prebiotics. The term prebiotic indicates "a substrate that is selectively utilized by host microorganisms conferring a health benefit", following the definition agreed by The International Scientific Association for Probiotics and Prebiotics (ISAPP) with a consensus statement [4]. At the moment, several molecules are recognized to correspond to this definition, and more are under investigation [5–7]. Several but not all prebiotics or candidate prebiotics are also dietary fiber. Without a doubt, fructans are among the most effective prebiotics and are also dietary fiber. Chicory is the prevalent source of purified fructans for the market [8]. In addition to fructans, other components of the root tissue, such as the cell wall components and various phytochemicals, contribute to the nutritional value of this source of prebiotics [8,9]. While there is a growing consensus concerning the relevance of an adequate intake of prebiotic for human and animal health, recent work has focused on the importance of using full plant tissues instead of the isolated prebiotic molecules, both in terms of positive health effects and food cycle sustainability [8,9]. Food containing a high quantity of fructans in addition to dietary fiber and other health-promoting phytochemicals can be a relevant functional food to implement

the diet of populations that consume low dietary fiber and prebiotics. An emerging body of evidence indicates that prebiotics can also alleviate the effects of several health threats such as those occurring under malnutrition, dysbiosis, or even in space missions [10]. In view of the recognized and growing evidence on the health benefits that nutritional fiber and prebiotic consumption can exert on human health, it is surprising that the use of chicory taproot as fresh vegetable is so limited. Chicory roots for industrial inulin extraction are produced in open field conditions from early summer to winter. Roots are harvested from October until February, reaching a total yield of fresh roots up to 50 t ha$^{-1}$ and an inulin content as high as 15%. Several cultivars are available on the market [11], and there is a large body of literature on agronomical and eco-physiological aspect of chicory cultivation for inulin production in open field conditions [12–14]. Research results demonstrated that inulin accumulation in root tissues can have a great variability under open field conditions due to the unpredictable change in environmental factors [15,16]. With respect to the traditional cultivation in open field conditions, controlled environment offers the opportunity to manage the environmental factors, to obtain faster crop growth, year-round production, and high-quality standard of vegetables, as reported for example for artichoke [17]. Limited studies are available on the production of fresh chicory roots under controlled conditions.

Plant Biostimulants (PBs), such as natural substances and beneficial microorganisms, could be an interesting tool to improve productivity and quality of vegetables in a sustainable manner, by reducing agrochemical inputs and preventing abiotic stress due to being out of season or intensive production systems. According to the EU Fertilising Products Regulation (2019/1009), PBs are defined on the basis of their agricultural functions for the improvement of one or more of the following characteristics: (1) nutrient use efficiency, (2) crop tolerance to abiotic stress, (3) quality traits, or (4) availability of confined nutrients in the soil or rhizosphere. PB's have been specifically designed for seed treatment, foliar spray, or soil drench application. Foliar spray is gaining significant importance in the market owing to its advantages, including better convenience in terms of application and better absorption than root drench application. Among biostimulant substances, protein hydrolysates (PHs) and seaweed extracts (SWE) have been successfully used as foliar sprays in many horticultural crops [18]. Several authors showed that PHs and SWE were able to stimulate plant primary and secondary metabolisms, resulting in enhanced growth and production as well as higher nutritional quality of horticultural crops [19–23]. Rouphael and coworkers [21] demonstrated that PHs and SWE treatment significantly increase the shoot fresh weight of lettuce plants, in both non-saline and salinity conditions. The same author [21] demonstrated that PHs and SWE application increase yield and dry matters as well as total phenolics and ascorbate content in spinach leaves. According to the results of Choi et al. (2022) [22], foliar and soil treatments of PHs enhanced shoot fresh weight and dry matters of romaine lettuce as well as fruit fresh weight, fruit number, and dry matter of tomato plants. The same work [22] showed that PHs enhanced chlorophyll and carotenoid content, antioxidant activity, and total phenolics of both studied species. Hussain et al. (2021) [23] showed that SWE had a positive effect on tomato yield and the total soluble solid content in tomato fruit, increasing the number of flower clusters and flowers, fruit number, dry weight of leaves, and roots. Scientific literature regarding the use of plant biostimulants is constantly increasing and involving different species and growing conditions. However, to our knowledge, there is no information about the potential benefits of using PBs on chicory for enhancing root yield and the accumulation of fructans.

The aims of this study were (i) to assess the feasibility of a controlled environment agriculture (CEA) system for Earth and space applications for chicory production in short and out-of-season cycles, with a variable level of environmental control and foliar applications of biostimulants to produce prebiotic rich roots for fresh consumption; (ii) to ascertain to which extent variable environmental control level can affect the prebiotic content of the fresh chicory root.

## 2. Materials and Methods

### 2.1. Greenhouse Trial

The trial was carried out from 14 December 2020 to 14 April 2021 in a polyethylene unheated greenhouse at the Experimental Farm of Tuscia University (latitude 42°25′ N, longitude 12°08′ E, altitude 310 m). Chicory (*Cichorium intybus* L. var. Bischoff) seeds from L'Ortolano srl (Cesena, Italy) were used as plant material. Plants were grown under natural light conditions and air temperature ranged from 7 to 27 °C. The soil was a sandy loam soil having a pH of 7.3, organic matter content of 18.8 g/kg, total nitrogen of 1.16 g/kg, assimilable P of 35 mg/kg, and cation exchange capacity of 25.3 meq/100 g. Cation exchange capacity was saturated with the following elements: Ca 67.7%, Mg 13.9%, K 9.8%, Na 8.6%. Treatments were three commercial biostimulant products and an untreated control. The tested products were a vegetal-derived protein hydrolysate (Trainer®—Hello Nature Italia srl, Rivoli Veronese, Italy; VPH), an animal-derived protein hydrolysate (Siapton®—Isagro S.p.A., Milan, Italy; APH), and a seaweed extract (Toggle®—Acadian Plant Health, Nova Scotia, Canada; SWE). The commercial biostimulant Trainer® was a legume-derived protein hydrolysate produced through enzymatic hydrolysis; it contained 190 g/kg of organic carbon and 50 g/kg of organic nitrogen, 310 g/kg of peptides, small amount of free amino acids, and no phytohormones. The commercial Siapton® was a collagen-derived protein hydrolysate produced in high pressure and thermal conditions (140 °C for 30 min at 3.6 atm). It contained 250 g/kg of organic carbon and 85 g/kg of organic nitrogen, 543.5 g/kg of total amino acids, and 79 g/kg of free amino acids. The detailed compositions of Trainer® and Siapton® were reported by Rouphael et al. (2021) [24]. Toggle® was derived from marine plant extract obtained by extraction with water solution containing acids or alkali. It contained 20 g/kg of organic carbon and 7 g/kg of mannitol. Biostimulant treatments were foliarly applied from 19 January every week until the end of the experiment for a total of 8 applications. Every foliar spray treatment was carried out with a 3 L spray bottle. Biostimulant concentrations (2.5 mL/L of VPH, 1.5 mL/L for APH, and 1.5 mL/L for SWE) used in each foliar application were those suggested in the product labels.

Treatments were arranged in a completely randomized block design with 4 replicates for a total of 16 plots having a surface of 2.5 m$^2$ each. Before sowing, soil was broadcast fertilized with 1.0 t ha$^{-1}$ of a granular mineral fertilizer NPK 12-12-17 (Blue Altea®—Hello Nature Italia srl, Rivoli Veronese, Italy). Chicory seeds were sown on 14 December 2020 in single rows spaced 20 cm apart. After emergence, seedlings were thinned at 10 cm in order to reach a plant density of 30 plants m$^2$. A drip irrigation system was set up between the rows in order to irrigate two rows per drip line. Drip lines had in-line emitters located 0.30 m lines apart and an emitter flow rate of 3.4 L h$^{-1}$. Weeds were removed by hand hoeing as necessary during the whole experiment.

In each biostimulant application date, the soil plant analysis development (SPAD) index and modulated chlorophyll fluorescence were measured. SPAD index was measured on fully expanded leaves by means of a portable chlorophyll meter SPAD-502 (Konica Minolta, Tokyo, Japan). Thirty healthy and fully expanded leaves were randomly measured and averaged to a single SPAD value for each experimental plot. Modulated chlorophyll fluorescence was measured, in dark adapted (for at least 15 min) leaves on three plants per plot, using a chlorophyll fluorometer Handy PEA (Hansatech Instruments Ltd., King's Lynn, UK) with an excitation source intensity higher than 3000 μmol m$^{-2}$ s$^{-1}$ at the sample surface. The minimal fluorescence intensity (F0) in a dark-adapted state was measured in the presence of a background far-red light to favor rapid oxidation of intersystem electron carriers. The maximal fluorescence intensities in the dark-adapted state (Fm) were measured by 0.8 s saturating pulses (3000 μmol m$^{-2}$ s$^{-1}$). The maximum quantum yield of the open photosystem II (PSII) (Fv/Fm) was calculated as (Fm–F0)/Fm.

Four plants per plot were sampled at 37 and 66 days after emergence (DAE) to weight the roots and calculate the average growth rate of the fresh biomass. Final harvest was performed 14 April 2021 (113 DAE). Forty plants per plot were harvested and leaves were

cut from the taproot in order to weigh them separately and determine the fresh weight of aerial part and roots. Leaves and roots were separately dried in a ventilated oven at 75 °C and in a freeze dryer, respectively, until constant weight to determine the dry weights. Inulin yield per hectare was calculated by multiplying the inulin concentration by root dry biomass and plant density.

### 2.2. Growth Chamber Trial

The plant material was the same used for the greenhouse experiment. Plants were grown at the CNR-IRET, under fully controlled environmental conditions in a growth chamber (Fitotron SGD170 Sanyo Gallenkamp, Birmingham, UK) [25]. The chamber was equipped with two LED lamps (model LX60, Heliospectra AB, Goteborg, Sweden). During the experiment, temperature was set at 20 °C during the day and 15 °C during the night, $CO_2$ was maintained at 400 ppm, relative humidity at 70%, and photoperiod was 12 h day/12 h night. For germination, two seeds per cell were seeded in a polystyrene seed tray and irrigated with distilled water, under 150 µmol $m^{-2}$ $s^{-1}$ photosynthetically photon flux density (PPFD), provided by cool white 5700 K LED. Germination occurred in six days; thereafter, the seedlings were irrigated with a nutrient solution having a pH of 6.5 and EC of 1.7 mS $cm^{-1}$ as the one selected for leafy plants in the EDEN ISS project [26]. Seedlings were grown in plastic pots (9 × 9 × 20 cm) filled with perlite and irrigated with the same nutrient solution. After the transplant, light intensity was increased to 550 µmol $m^{-2}$ $s^{-1}$ PPFD, with a combination of cool white 5700 K, red, and blue LEDs. Plant density in the chamber was set at 16 plant $m^{-2}$.

Chicory plants cultivated in the growth chamber were foliar treated with the two plant biostimulants used in the greenhouse trial: Trainer[®] (VPH) and Toggle[®] (SWE); the tested rates were those used in the greenhouse trial (2.5 mL/L for VPH, and 1.5 mL/L for SWE). Due to the negative performances in the greenhouse trial, the APH (Siapton[®]) was not used in the growth chamber experiment. A control treatment where leaves were sprayed with only water was also included. Foliar treatments started twenty-eight DAE and were carried out every week, until the end of the experiment for a total of four applications. The three treatments were arranged in a completely randomized design with eighteen plants per treatments.

At 30, 37, and 55 days after planting (DAP), four plants per plot were collected and weighted to determine the fresh biomass. At final harvest, six plants per treatment were collected and the roots were quickly washed, then the leaves were cut from the taproots and the fresh weight of the two organs was measured. The same samples used for the fresh biomass analysis were dried with a freeze dryer in order to assay the dry matter content and the non-structural carbohydrate content.

### 2.3. Non-Structural Carbohydrate

For compositional analysis, dried roots from both trials were milled to a fine powder using an MF 10 miller IKA (IKA[®]-Werke GmbH & Co. KG, Staufen, Germany) and sieved to screen particles until a 0.5 mm grid. For non-structural carbohydrate (NSC) extraction, 20 mg of dried root powder was quenched in 0.5 mL of 100% ethanol at 70 °C in open screw plastic tubes until complete ethanol evaporation was achieved. The solid residue was resuspended in 1 mL of water and extracted at 70 °C for 2 h under vigorous stirring. The extract was centrifuged at 9000× *g* for 10 min. The supernatant was passed through a nylon filter 0.45 µm PPII syringe filters (Whatman Inc., Maidstone, UK) and subsampled. One aliquot was used for fructans hydrolysis [27] and a second aliquot for the measurements of glucose, fructose, and sucrose. Fructans were hydrolyzed in 60 mM HCL for 2 h at 70 °C with vigorous stirring. Glucose, fructose, and sucrose present in the first aliquot of the extract and fructose and glucose derived by the fructans hydrolysis were measured by high-performance anion exchange chromatography, with pulsed amperometric detection (HPAEC-PAD), equipped with a gold working electrode (1.0 mm in diameter) and an Ag/AgCl reference electrode (Dionex™ ICS-5000, Thermo Fisher Scientific, Waltham, MA,

USA). An analytical CarboPac PA100 column (4 mm × 250 mm) with a relative guard column was used. All runs were carried out at 30 °C using a mobile phase gradient with two aqueous solutions: (A) NaOH 1 mol L$^{-1}$; (B) Na-acetate 1 mol L$^{-1}$ at a flow rate 1 mL min$^{-1}$ as described in [28]. Calculations of inulin concentration and average degree of polymerization ($DP_{av}$) is as reported in Verspreet et al. [29]:

$$Inulin\ (\%) = k\left(G_f + F_f\right) \tag{1}$$

$$k = \frac{180 + 162(DP_{av} - 1)}{180DP_{av}} \tag{2}$$

$$DP_{av} = \frac{F_f}{G_f} + 1 \tag{3}$$

where $G_f$ and $F_f$ are the concentration of glucose and fructose released from fructan and are calculated as follow:

$$G_f\ (\%) = \frac{180.16V_{ex}(G_a - G_b - S_a)}{10000m_s} \tag{4}$$

$$F_f\ (\%) = \frac{180.16V_{ex}(F_a - F_b - S_a)}{10000m_s} \tag{5}$$

where 180.16 is the molecular weight of glucose or fructose, $V_{ex}$ is the volume of the extract (mL), $m_s$ is the sample mass (mg), $G_a$ and $G_b$ are the concentration of glucose (μM) in the nonhydrolyzed and hydrolyzed sample, respectively, $F_a$ and $F_b$ are the concentration of fructose in the nonhydrolyzed and hydrolyzed sample, respectively (μM), and $S_a$ is the concentration of sucrose in the nonhydrolyzed sample (μM).

### 2.4. Statistical Analysis

The statistical analysis was carried out using IBM SPSS Statistics 20 (Chicago, IL, USA). All the parameters of the greenhouse and growth room experiments were subjected to one-way analysis of variance (ANOVA). Mean values were separated according to Tukey's test with $p = 0.05$.

## 3. Results

### 3.1. Greenhouse Experiment

In the greenhouse experiment, seedling emergence occurred nine days after sowing (DAS) and the first true leaf stage was recognized 18 days after emergence (DAE). The start of root thickening was visually observed at 66 DAE when the root fresh weight resulted in an average of 10.5 g. The average growth rate of fresh root biomass from 37 to 66 DAE and from 66 DAE until the end of the experiment (last 47 days), resulted in 0.37 g·day$^{-1}$ and 1.60 g·day$^{-1}$, respectively. Bolting rate at harvest was not significantly affected by treatments (avg. 51%). Treatments did not significantly affect the SPAD index and maximum quantum yield of PSII of chicory leaves, as shown in Tables 1 and 2.

**Table 1.** Effect of biostimulant treatments on SPAD index of chicory leaves during the greenhouse trial.

| Treatment | SPAD Index | | | | | | | |
|---|---|---|---|---|---|---|---|---|
| | **66 DAS** | **73 DAS** | **80 DAS** | **87 DAS** | **94 DAS** | **101 DAS** | **108 DAS** | **115 DAS** |
| Control | 35.3 | 43.3 | 38.0 | 39.9 | 42.4 | 45.7 | 49.5 | 76.0 |
| VPH | 36.4 | 41.5 | 41.5 | 40.6 | 42.9 | 46.6 | 48.5 | 75.2 |
| APH | 35.8 | 40.4 | 42.3 | 39.8 | 38.4 | 40.9 | 46.0 | 76.8 |
| SWE | 35.4 | 41.7 | 39.5 | 41.2 | 40.2 | 42.7 | 47.6 | 68.3 |
| Significance | ns | ns | ns | ns | ns | ns | ns | ns |

VPH = Vegetal Protein Hydrolysate; APH = Animal Protein Hydrolysate; SWE = Seaweed Extract. DAS = Days after sowing. ns = not significant.

**Table 2.** Effect of biostimulant treatments on maximum quantum use efficiency of PSII (Fv/Fm) of chicory leaves during the greenhouse trial.

| Treatment | Fv/Fm | | | | | | | |
|---|---|---|---|---|---|---|---|---|
| | 66 DAS | 73 DAS | 80 DAS | 87 DAS | 94 DAS | 101 DAS | 108 DAS | 115 DAS |
| Control | 0.805 | 0.772 | 0.790 | 0.807 | 0.791 | 0.794 | 0.820 | 0.793 |
| VPH | 0.814 | 0.762 | 0.778 | 0.815 | 0.769 | 0.813 | 0.818 | 0.783 |
| APH | 0.811 | 0.769 | 0.783 | 0.788 | 0.760 | 0.789 | 0.798 | 0.800 |
| SWE | 0.813 | 0.772 | 0.789 | 0.805 | 0.792 | 0.805 | 0.797 | 0.783 |
| Significance | ns | ns | ns | ns | ns | ns | ns | ns |

VPH = Vegetal Protein Hydrolysate; APH = Animal Protein Hydrolysate; SWE = Seaweed Extract. DAS = Days after sowing. ns = not significant.

The maximum quantum yield of PSII ratio (Fv/Fm) of leaves has been largely used as a sensitive marker of plant photosynthetic performance; Fv/Fm significantly decreased in leaves when PSII photoinhibition is induced by stress conditions. In the greenhouse trial, the Fv/Fm values of chicory leaves, displayed in Table 2, were always in the range typically recorded for healthy plants (0.75–0.85).

Table 3 shows the results of plant biomass analysis and inulin content in chicory roots, while the concentrations of glucose, fructose, and sucrose in chicory roots are reported in Table 4. VPH applications determined a statistically significant increase of fresh and dry weight, and diameter of taproot in comparison with the application of APH while SWE and Control treatments gave intermediate values which were not statistically different from both VPH and APH treatments (Table 3). No statistical differences were recorded for leaf fresh and dry biomass, leaf dry matter, root length, root dry matter, shoot/root ratio, and inulin concentration in roots (Table 3). The degree of polymerization of inulin was on average 9.17%, and no differences resulted from the biostimulant treatments (data not shown). Average leaves and root fresh weight of the whole trial were 130 g·plant$^{-1}$ and 42 g·plant$^{-1}$, respectively, corresponding to a production of fresh leaf and root of 38.98 t·ha$^{-1}$ and 12.59 t·ha$^{-1}$, respectively. The inulin yield per hectare (Figure 1), was significantly increased by VPH (avg + 29%) compared to Control and APH treatments. The SWE treatment resulted in an intermediate value with no significant differences with VPH, APH, and Control treatment.

**Table 3.** Effects of plant biostimulants on fresh and dry biomass of leaves and roots, root characteristics, shoot to root ratio, and root inulin concentration in the greenhouse chicory trial.

| Treatment | Leaf Fresh Weight | Leaf Dry Matter | Leaf Dry Weight | Root Fresh Weight | Maximum Root Diameter | Root Length | Root Dry Matter | Root Dry Weight | Shoot/Root | Inulin Concentration |
|---|---|---|---|---|---|---|---|---|---|---|
| | (g plant$^{-1}$) | (%) | (g plant$^{-1}$) | (g plant$^{-1}$) | (cm root$^{-1}$) | (cm root$^{-1}$) | (%) | (g plant$^{-1}$) | | (% dry wt.) |
| Control | 134.7 | 5.89 | 8.33 | 42.1 ab | 2.19 ab | 17.0 | 15.3 | 6.44 ab | 1.65 | 44.7 |
| VPH | 139.4 | 6.10 | 10.02 | 48.6 a | 2.37 a | 18.0 | 15.8 | 7.98 a | 1.65 | 46.2 |
| APH | 117.0 | 6.44 | 10.65 | 35.4 b | 2.05 b | 16.4 | 17.0 | 6.02 b | 1.94 | 47.4 |
| SWE | 128.7 | 6.08 | 10.67 | 41.9 ab | 2.29 ab | 17.0 | 17.1 | 7.18 ab | 1.74 | 47.9 |
| Significance | ns | ns | ns | ** | * | ns | ns | * | ns | ns |

VPH = Vegetal Protein Hydrolysate; APH = Animal Protein Hydrolysate; SWE = Seaweed Extract. *, ** Significant at $p < 0.05$ or 0.01. ns = not significant. Different letters within each column indicate significant differences according to Tukey's multiple-range test ($p = 0.05$).

The average concentration of glucose, fructose and sucrose in chicory roots were 1.21%, 0.33%, and 2.78%, respectively (Table 4).

**Table 4.** Effect of biostimulant treatments on glucose, fructose, and sucrose concentration of chicory roots in the greenhouse trial.

| Treatment | Soluble Carbohydrate Concentration (% dry wt.) | | |
|---|---|---|---|
| | Glucose | Fructose | Sucrose |
| Control | 1.18 | 0.33 | 3.00 |
| VPH | 1.38 | 0.30 | 2.59 |
| APH | 1.37 | 0.32 | 3.10 |
| SWE | 0.92 | 0.37 | 2.45 |
| Significance | ns | ns | ns |

VPH = Vegetal Protein Hydrolysate; APH = Animal Protein Hydrolysate; SWE = Seaweed Extract. ns = not significant.

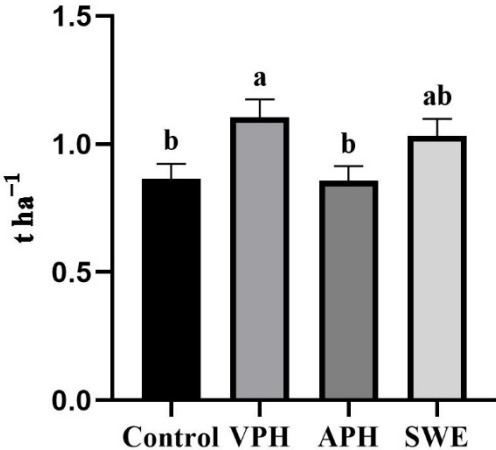

**Figure 1.** Effect of biostimulants on inulin yield per hectare in greenhouse chicory. Bars indicate the standard errors of the mean values. Different letters indicate significant differences according to Tukey's multiple-range test ($p = 0.05$). VPH = Vegetal Protein Hydrolysate; APH = Animal Protein Hydrolysate; SWE = Seaweed Extract.

### 3.2. Growth Chamber Experiment

In the growth chamber harvest occurred 64 days after planting (DAP). At 30, 37, 55 DAP the average weights for fresh taproots were 8.80, 21.57, and 42.24 g, respectively. The taproot biomass growth rates between 0–37 and 37–64 DAP were not statistically different between treatments and resulted in an average of 0.58 and 0.81 g·day$^{-1}$, respectively. At harvest, no bolting was detected in all chicory plants. No statistical differences were determined by the biostimulant treatments on the selected plant variables (Tables 5 and 6). In terms of inulin yield, on the hectare basis, the treatments showed no significant differences in comparison with untreated plants, resulting in 1.0 t·ha$^{-1}$ for VPH and SWE and 0.7 t·ha$^{-1}$ for the control. The estimated average production resulted in 12.2 t·ha$^{-1}$ and 6.95 t·ha$^{-1}$ of fresh leaf and root, respectively. Taproot fresh weight per plant was 43.5 g·plant$^{-1}$ on average. Average leaf fresh weight was 76.78 g·plant$^{-1}$. The average percentage of taproot dry matters at the end of the experiments was 25.5%. The related dry mass production amounted to 1.8 t·ha$^{-1}$. No substantial differences were found for S/R ratio that resulted in an average of 0.98. Average inulin content was 57% of the dry matter. Thus, average inulin production resulted in 0.95 t·ha$^{-1}$. The average degree of polymerization of the inulin chain resulted in an average of 13.5% with no significant differences among treatments (data not shown).

**Table 5.** Effects of plant biostimulants on fresh and dry biomass of leaves and roots, root characteristics, shoot to root ratio, and root inulin concentration in growth chamber chicory trial.

| Treatment | Leaf Fresh Weight | Leaf Dry Matter | Leaf Dry Weight | Specific Leaf Dry Weight | Root Fresh Weight | Root Dry Matter | Root Dry Weight | Shoot/Root | Inulin Concentration |
|---|---|---|---|---|---|---|---|---|---|
| | (g plant$^{-1}$) | (%) | (g plant$^{-1}$) | (g cm$^{-2}$) | (g plant$^{-1}$) | (%) | (g plant$^{-1}$) | | (% dry wt.) |
| Control | 67.63 | 12.20 | 10.45 | 2.93 | 37.63 | 25.05 | 9.45 | 1.11 | 54.42 |
| VPH | 87.55 | 12.74 | 12.74 | 2.82 | 46.73 | 27.02 | 12.84 | 0.99 | 59.57 |
| SWE | 73.70 | 10.76 | 9.65 | 3.09 | 46.00 | 24.69 | 11.40 | 0.85 | 56.65 |
| Significance | ns | ns | ns | ns | ns | ns | ns | ns | ns |

VPH = Vegetal Protein Hydrolysate; SWE = Seaweed Extract. ns = not significant.

**Table 6.** Effect of biostimulant treatments on glucose, fructose, and sucrose concentration of chicory roots in the growth chamber trial.

| Treatment | Soluble Carbohydrate Concentration (% dry wt.) | | |
|---|---|---|---|
| | Glucose | Fructose | Sucrose |
| Control | 0.89 | 0.83 | 1.63 |
| VPH | 1.09 | 1.33 | 1.41 |
| SWE | 0.92 | 0.89 | 1.73 |
| Significance | ns | ns | ns |

VPH = Vegetal Protein Hydrolysate; SWE = Seaweed Extract. ns = not significant.

The average concentration of glucose, fructose and sucrose in chicory roots were 0.97%, 1.02%, and 1.59%, respectively (Table 6).

## 4. Discussion

Controlled environments (full or semi) allow to grow chicory for out-of-season production of health promoting vegetables, with high prebiotic potential. Different works give insights regarding the full field cultivation aimed to the conventional production of inulin, extracted, and used for food and industrial purposes [13,15,30]. Less is known about out-of-season cultivation conditions oriented to the production of fresh vegetables and ready to eat products. The choice of suitable environmental and agronomic inputs, in any case, is essential for controlling the production cycle and guaranteeing the quality and quantity of the yield.

For chicory taproot, Van Arkel and coworkers [13] described three growing phases based on changes observed in the pattern of inulin accumulation in field conditions. These phases can be well correlated with the phenological status of the crop and are influenced by environmental factors: phase 1—onset of root thickening and inulin biosynthesis; phase 2—substantial increase of root mass and inulin accumulation but decreasing of the inulin polymerization grade; phase 3—polymerization grade continues to decrease and the yield remains constant. In our experiments, only phase 1 occurred due to the short cycle adopted. This phase in Van Arkel [13] field condition experiment occurred in one month. According to the growth rate of our experiments, the start of root thickening and the onset of phase 1 occurred in two months for greenhouse and one month in the growth chambers determining different duration of the two experiments. In the greenhouse, the delay of the start or root thickening could be attributable to the low temperature and low light availability of the winter period. In our experiment, sowing time (held in December) largely anticipates the conventional growing season of chicory taproot, that usually starts in early summer in full field conditions. On the other hands, in the growth chamber, thanks to full environmental control, plants experienced constant and largely optimal conditions both in terms of temperature and light availability during the full growth cycle, resulting in a faster growth rate.

The sowing time in the greenhouse and the management of environmental parameters in the growth chamber resulted also to be an important factor that could influence the tran-

sition to the reproductive stage in chicory plant. Despite the fact that chicory inflorescence has high feed quality and is desirable for foraging production [31,32], in chicory cultivation for root production, bolting is a negative occurrence since it causes lignification of the root, a decrease in root productivity, and might allow seed dispersal, favoring the spread of the species as a weed [33]. Floral induction is mainly driven by low temperature, and flowering requires a long day photoperiod [34]. However, cold requirement depends on the variety, age of the plant, and occurrence of other environmental stresses; high temperature, for example, can devernalize chicory or induce bolting depending on concurring environmental and physiological variables [14]. In the unheated greenhouse, low temperature in early phases led to floral induction and bolting occurred under favorable photoperiod. Moreover, vernalization conditions are unfavorable for a rapid plant growth. In the greenhouse trial, low temperatures up to 7 °C induced vernalization, and fast stem elongation started at the beginning of April, in concomitance of the occurrence of longer photoperiod and high temperatures (up to 27 °C). Hence, greenhouse experiment's growing season resulted to be favorable for chicory flower induction and bolting. No bolting occurred in the growth room since temperatures were higher, allowing a faster development of plants while avoiding vernalization. Reduced bolting can be obtained by genetic improvement and by a high level of temperature control in the greenhouse. Less feasible, although technically possible, appears to be the control of the photoperiod. A few works have shown a correlation between the use of plant biostimulants and early flowering [35,36] and an increase of flower number [37] in horticultural crops, but no information is reported for chicory. In our trials, we did not observe any effects of tested biostimulants on bolting of chicory plants.

The management of variables related to light Input, such as light intensity, photoperiod, and DLI, is a powerful tool to modulate fresh and dry biomass, growth rate, along with the duration of the production cycle of plants. Moreover, especially in growth chambers, a correct use of light inputs results in high use efficiency of resources such as water, light, and energy (WUE, LUE, and EUE), essentials for a sustainable and feasible indoor cultivation. Pennisi et al. (2020) [38] compared three different DLI (14.4, 18, and 21.6 $\mathrm{mol \cdot m^{-2} \cdot d^{-1}}$) in chicory cultivar. Plants grown under 14.4 $\mathrm{mol \cdot m^{-2} \cdot d^{-1}}$ resulted in higher leaf area (+81%), leaf fresh biomass (+47%), along with high WUE (33%), LUE (57%), and EUE (120%) when compared to a DLI of 21.4 $\mathrm{mol \cdot m^{-2} \cdot d^{-1}}$. The fresh biomass and the other use efficiency parameters decrease as the DLI increases. In contrast, plants grown with DLI of 21.4 $\mathrm{mol \cdot m^{-2} \cdot d^{-1}}$ resulted in higher leaf dry matter content. The increase in dry matter content in plants undergoing longer days was previously related to higher carbohydrate production through photosynthesis in lettuce [39]. An increase in photosynthesis due to the optimal light management can result in an increase of photosynthate allocation in sink organs [25]. In chicory taproot plants, an increase of the source potential due to high DLI could hypothetically result in higher inulin yield and DP in roots especially in a short production cycle when the root is harvested during an ongoing active accumulation phase. Our results indicate that in the growth chamber, where the DLI was 23.76 $\mathrm{mol \cdot m^{-2} \cdot d^{-1}}$, leaf and taproot dry matter and inulin content resulted higher than in the greenhouse where the estimated average DLI was 15 $\mathrm{mol \cdot m^{-2} \cdot d^{-1}}$, although no considerable differences were detected for the single root weight. By contrast, leaf fresh biomass resulted higher in greenhouse than in the growth chamber. Taproot fresh biomass production per unit area of the greenhouse resulted to be 2.3-fold higher than in the growth chamber, but this was associated with the higher greenhouse plants' density. However, dry matter content in the growth chamber showed a 1.6-fold increase with respect to the greenhouse chicory, resulting in almost the same inulin yield per square meter. The higher inulin DP obtained in the growth chamber could be attributed to the optimal range of temperature adopted. Low temperatures increase the effect of the hydrolytic activity of fructan exohydrolases (FEH) enzymes, promoting fructans depolymerizations [40]. In the growth room, temperatures were kept at 20 °C during the day and 15 °C at night, and at this temperature range a low FEH activity is expected. This consideration suggests the possibility to modulate the DP in order to achieve the desired chain length in observation of the market objectives [41].

Moreover, growth room production gave a high amount of fresh leaf and taproot biomass. The grow room data demonstrate that a fully controlled environment allows to shorten the production cycle and to better control quality attributes of chicory taproots. Cultivation in vertical farming systems could represent a feasible and sustainable manner to produce chicory in urban areas coupling the increase of the cultivation surface allowed by vertical farming [42] with the possibility to control production calendar, produce availability, and its quality characteristics.

Plant biostimulants, showed the potential to affect differentially the crop performances from quali-quantitative point of view. The greenhouse production of inulin calculated on the hectare basis resulted to be statistically higher by 29% in the VPH treatment in comparison with Control and APH treatments. The above findings are mainly related to the significant positive effects of biostimulant on root dry matter production. Colla and coworkers [19] demonstrated that bioactive compounds in VPH such as small peptides and amino acids can stimulate plant growth and productivity through the modulation of primary and secondary metabolism. Moreover, several bioassays reported that the tested VPH 'Trainer$^{®}$' exerted an auxin-line activity stimulating root growth [43,44]. On the other hand, APH treatment caused a decrease of (i) the root fresh weight, (ii) the maximum root diameter, (iii) the root dry weight, and iv) the fresh and dry root production per hectare with respect to the application of VPH. Phytotoxicity effects along with suppression of growth in concomitance with the use of animal-derived PHs were already reported by several authors [24,45–47]. The phenomenon is recognized as "general amino acid inhibition", and it is due to an excessive leaf uptake of free amino acids leading to intracellular amino acid imbalance [48]. Based on the negative performances of APH under greenhouse trial, only VPH and SWE were considered in the growth chamber experiment. The lack of significant differences among biostimulant treatments under growth chamber conditions is in line with the role of biostimulants to ameliorate plant productivity under limiting growing conditions. In the growth chamber, plants experienced optimal growing conditions, reducing the benefits of biostimulant applications. However, it should also be considered that the small sample size under growth chamber conditions together with the high genetic variability of the chicory open-pollinated cultivar increased the experimental error making difficult to detect significant differences among biostimulant treatments.

Inulin yield is considered the most important criterion in taproot chicory production and breeding [49]. In a conventional spring to autumn production cycle, recorded inulin yields per hectare are ten-fold higher than the one obtained here under different types of controlled conditions [50]. However, conventional productions are aimed to the extraction and isolation of the polymer in order to satisfy food and industrial market requests [13]. Our work was focalized on the production of prebiotics for rich, fresh vegetables throughout a fast and out-of-season production cycle, and we demonstrated the feasibility of this production procedure. The taproot at this production stage can be eaten fresh or cooked, or easily transformed by the food industry to be consumed in various kinds of healthy, ready-to-eat products. The young taproots, harvested much early than in normal field production systems, were rich in prebiotics and valuable for the fresh products market. Furthermore, when different levels of environmental control were applied, we obtained fresh roots of similar size, a couple of which could be considered a reasonable food serving, but with different inulin content. The roots harvested in the growth room trial had higher inulin content than the roots produced under greenhouse conditions. A serving of 100 g of chicory root produced in the growth chamber could provide on average about 14.5 g of inulin, while the same amount of root obtained in the greenhouse would provide almost half of that amount.

There is a fast-rising body of evidence indicating that the daily consumption of prebiotics in addition to fiber can be a powerful option to ameliorate human and animal health [51–53]. According to EFSA, the recommended daily intake of "chicory-type inulin" for human is up to 12 g/day [54]. Recently, it was pointed out that eating the full chicory root can be a better way of prebiotic intake than assuming inulin-based nutritional

integrators where inulin is almost pure or at very high concentration [8] since the root also contains cell wall polymers, such as cellulose, hemicellulose, pectin, and lignin, and other phytochemicals. An excessive intake of inulin and fiber can cause discomfort or have detrimental effects on gut functioning; hence, an inulin-rich food should be managed easily to provide the correct amount of prebiotic. Our results on the differences among growth chamber- and greenhouse-produced chicory taproot show that the inulin amount in 100 g of fresh root can be modulated within a large interval via environmental control of the growing conditions. This opens new and interesting perspectives for developing controlled agriculture systems for Earth- and space-oriented functional food production.

## 5. Conclusions

The results demonstrated the possibility to produce fresh chicory taproot with high prebiotic content in a short growing cycle and out of season both in greenhouse and growth chambers. The controlled environment offers the opportunity to modulate the growing conditions to optimize the inulin yield. The application of plant biostimulants showed significant effects only under greenhouse conditions with an increase of inulin yield by 27.8% with foliar sprays of VPH compared with untreated control. The foliar applications of the other two biostimulants (APH and SWE) were not effective to enhance crop growth and inulin yield in comparison with untreated control. In conclusion, the trials also highlighted that controlled-environment agriculture is a suitable way of producing young chicory taproot for fresh consumption and highlighted the potential benefits of this production system in modulating prebiotic content of chicory roots for Earth and space applications. Finally, further studies could identify the potential benefits of PBs for seed treatment or root application and to understand the mode of action of PBs through metabolomics.

**Author Contributions:** Conceptualization, A.B. and G.C.; methodology, S.P., M.C., S.M., A.B. and G.C.; validation, G.P., S.P., M.C. and S.M.; formal analysis, G.P., S.P., S.M. and M.C.; investigation, G.P., S.P., M.C. and S.M.; resources, G.C. and A.B.; data curation, G.P., S.P. and M.C.; writing—original draft preparation, G.P.; writing—review and editing, S.P., M.C., S.M., G.C. and A.B.; visualization, G.P., S.P., M.C. and S.M.; supervision, G.C. and A.B. project administration, G.C. and A.B.; funding acquisition, G.C. and A.B. All authors have read and agreed to the published version of the manuscript.

**Funding:** This work is part of the Ph.D. thesis of Gabriele Paglialunga at Tuscia University and it was partially funded by MIUR in the frame of the initiative "Departments of excellence", Law 232/2016 and by the Agenzia Spaziale Italiana (ASI) and the National Research Council, Research Institute on Terrestrial Ecosystems (CNR-IRET) in the frame of the Agreement: 2021-2-HH.0 "Sistemi e tecnologie per la produzione di microortaggi nello Spazio 'Microgreens x Microgravity' (MICROx2)". Support for this study was provided by the project: In situ Resource Bio-Utilization per il support alla vita nello Spazio (ReBUS) from the Agenzia Spaziale Italiana (ASI): Prot. ASI n. 1714 del 19 feb. 2018—n° DC-VUM-2017-080 Bando di Ricerca per missioni future di esplorazione umana dello spazio—Area tematica Sistemi Biorigenerativi.

**Data Availability Statement:** Not applicable.

**Acknowledgments:** The authors acknowledge the technical support of Antonio Fiorillo in the management of the greenhouse trial.

**Conflicts of Interest:** The authors declare no conflict of interest.

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
