# Peer review of "Chicory Taproot Production: Effects of Biostimulants under Partial or Full Controlled Environmental Conditions"

_agronomy, doi:10.3390/agronomy12112816_

Round 1

Reviewer 1 Report

This submission reports the success in the off-season production of chicory root with two systems: green house and growth chamber. The important growth parameters and important traits were closely monitored and assessed. Another important conclusion is the effectiveness of one vegetal-derived protein hydrolysate (VPH) in promoting greenhouse production with higher root fresh weight, dry weight, and higher inulin yield per hectare. The experiments were well designed, executed and presented. Overall, it is a good agronomy paper with general interests. 

Two points from this reviewer for the authors to address: 

1. In section 2,3 (Non-structural carbohydrate), inulin determination is not clearly elaborated. It would be good to add the formula for computing inulin values. 

2. The lack of significance in all the growth parameters and inulin concentration for growth chamber chicory is a little surprising. From table 2, multiple parameters for VPH treatment are >10% higher than the numbers for the control, e.g., leaf fresh weight, leaf dry weight, root fresh weight, root dry weight, and also the inulin concentration. With the assumption of higher uniformity in chamber growth plants, statistical significance is expected. The authors need to double-check the statistical analysis. If the same results are obtained, an explanation is needed for the lack of statistical significance. 

Author Response

This submission reports the success in the off-season production of chicory root with two systems: green house and growth chamber. The important growth parameters and important traits were closely monitored and assessed. Another important conclusion is the effectiveness of one vegetal-derived protein hydrolysate (VPH) in promoting greenhouse production with higher root fresh weight, dry weight, and higher inulin yield per hectare. The experiments were well designed, executed and presented. Overall, it is a good agronomy paper with general interests.

Dear Reviewer, thank for considering the manuscript well designed, executed and presented and suitable for the Journal.

Two points from this reviewer for the authors to address:

  1. In section 2,3 (Non-structural carbohydrate), inulin determination is not clearly elaborated. It would be good to add the formula for computing inulin values.

The formula for inulin determination and the respective literature reference have been inserted in the M&M section as requested.

  1. The lack of significance in all the growth parameters and inulin concentration for growth chamber chicory is a little surprising. From table 2, multiple parameters for VPH treatment are >10% higher than the numbers for the control, e.g., leaf fresh weight, leaf dry weight, root fresh weight, root dry weight, and also the inulin concentration. With the assumption of higher uniformity in chamber growth plants, statistical significance is expected. The authors need to double-check the statistical analysis. If the same results are obtained, an explanation is needed for the lack of statistical significance.

Dear Reviewer, let me draw your attention that open pollinated cultivars are used for commercial chicory root production leading to a great population-level genetic diversity. The high genetic diversity in commercial cultivars is also expressed in the plant's phenotype. Moreover, the number of sampled plants in the growth chamber trial was lower than in the greenhouse trial due to the limited space available in the growth chamber. The above findings (high genetic variability, low number of replicates) together with the optimal growing conditions under growth chamber may explain the lack of significance differences in the growth chamber experiment. 

Reviewer 2 Report

Several commercial, amino acid products of different origin have been evaluated in order to assess the possible biostimulant activity on chicory plants.

I consider that the article is interesting and provides new data about the effect of biostimulants on crops, in this case on enhancing compounds related with the prebiotic capacity of chicory root.

Author Response

Several commercial, amino acid products of different origin have been evaluated in order to assess the possible biostimulant activity on chicory plants.

I consider that the article is interesting and provides new data about the effect of biostimulants on crops, in this case on enhancing compounds related with the prebiotic capacity of chicory root.

Dear Reviewer, thank for considering the manuscript interesting and suitable for the Journal.

Reviewer 3 Report

The main purpose of the work is to analyze conditions that shorten and simplify the production of chicory for its use as a source of inulin. Results are interested; however the work does not present an in-depth study in this regard. So, in addition to foliar treatment, biostimulants should also be applied to the root. Moreover, parameters that can clarify the difference in the effect of the three biostimulants used, such as primary and secondary metabolites, should have been studied.

Additionally- although authors explain that ”In our experiments, only phase 1 occurred due to the short-cycle adopted. ..” ,  it would be quite interested to extend the study to phase 2 and 3 in order to search whether biostimulants have any influenced on extend or shorten the phases.

Finally in the manuscript there are some points that should be reviewed

Major point:

 Point 1- It should be useful to show a table with individual values, corresponding to each treatment, of soil plant analysis development (SPAD) index and chlorophyll fluorescence.

Point 2- according to materials & methods, non-structural carbohydrates were assayed (line 201: Glucose, fructose   and sucrose present in the first aliquot of the extract and fructose and glucose derived by  the fructans hydrolysis were measured by high-performance anion exchange chromatog- raphy, with pulsed amperometric detection….) however there are no data from them in results

Minor points-

Point 3-

Although line 389 (Discussion) explains the reason why only APH biostimulant was not used in Growth chamber experiments, it would be convenient if it were explained beforehand in the text, in results.

Point 4-

Line 111: … cation exchange capacity of 25.3 meq/100 g cation exchange capacity was saturated with the following elements….

It should be: … cation exchange capacity of 25.3 meq/100 g. Cation exchange capacity ….

Point 5-

Line 133: Before sowing, soil was broadcast fertilized with (1.0 t ha-1) of a granular …

Parentheses should be removed

Point 6- Line 185: DAT?

DAT must be defined

Author Response

The main purpose of the work is to analyze conditions that shorten and simplify the production of chicory for its use as a source of inulin. Results are interested; however the work does not present an in-depth study in this regard. So, in addition to foliar treatment, biostimulants should also be applied to the root. Moreover, parameters that can clarify the difference in the effect of the three biostimulants used, such as primary and secondary metabolites, should have been studied.

Additionally- although authors explain that ”In our experiments, only phase 1 occurred due to the short-cycle adopted. ..” ,  it would be quite interested to extend the study to phase 2 and 3 in order to search whether biostimulants have any influenced on extend or shorten the phases.

Finally in the manuscript there are some points that should be reviewed.

Dear Reviewer, we agree with you that this study represents a first relevant step to study environmental and agronomical control of chicory growth potential and inulin accumulation under controlled conditions. Biostimulant products were selected among those most used at commercial level and already successfully tested in several crops. The selected products are specifically designed for foliar applications while other products have been developed for root application. Moreover, foliar application is the preferred application method by farmers due to the fast crop response and the easy way of application. We would like to thank the Reviewer for suggesting further studies on evaluating the benefits of biostimulants for root application and to extend the study to phase 2 and 3. Finally, all points raised by Reviewer have been addressed as reported below.

Major point:

Point 1- It should be useful to show a table with individual values, corresponding to each treatment, of soil plant analysis development (SPAD) index and chlorophyll fluorescence.

Table with the SPAD index and chlorophyll fluorescence has been inserted as Supplementary material.

Point 2- according to materials & methods, non-structural carbohydrates were assayed (line 201: Glucose, fructose   and sucrose present in the first aliquot of the extract and fructose and glucose derived by  the fructans hydrolysis were measured by high-performance anion exchange chromatography, with pulsed amerometric detection….) however there are no data from them in results.

We added the contents of glucose, fructose and sucrose for both trials in supplementary material. In addition, please note that the formula for  determination of inulin has been added to M&M; the formula includes the single sugars as variables.  

Minor points-

Point 3-Although line 389 (Discussion) explains the reason why only APH biostimulant was not used in Growth chamber experiments, it would be convenient if it were explained beforehand in the text, in results.

We included in the revised version of the manuscript the reasons for the exclusion of APH in the growth chamber trial (see materials and methods section 2.2 line 181)

Point 4-Line 111: … cation exchange capacity of 25.3 meq/100 g cation exchange capacity was saturated with the following elements….It should be: … cation exchange capacity of 25.3 meq/100 g. Cation exchange capacity ….

The text has been corrected

Point 5- Line 133: Before sowing, soil was broadcast fertilized with (1.0 t ha-1) of a granular …

Parentheses should be removed

The text has been corrected.

Point 6- Line 185: DAT?

DAT must be defined

The correct abbreviation was reported in the text as DAP (days after planting).

Round 2

Reviewer 3 Report

Although some points have been addressed by authors, the work still lacks an in-depth study that supports the main goal of the work. As it was already suggested, addition to foliar treatment, biostimulants should also be applied to the root and analyze parameters that can clarify the difference in the effect of the three biostimulants used, such as primary and secondary metabolites, should be studied.

Author Response

Dear Reviewer, thank for your comments but as indicated in the previous discussion section we focus on foliar application of biostimulants because this is the most popular way to applied biostimulants at farm level. Moreover, the aim of the work was to present for the first time to the scientific community the potential benefits of applying biostimulants to chicory crop considering the effects on quali-quantitative traits.  To be clearer for the reader, we added in the introduction of revised manuscript a sentence clarifying that foliar application of plant biostimulants is the most popular application method (L78-81);  we also added in the conclusion a sentence reporting your suggestions for future investigations on other biostimulants for seed treatment or root application and metabolomic study (L446-448).

Round 3

Reviewer 3 Report

Although the authors have explained why they use only foliar application of biostimulants, I still consider that the work does not represent an in-depth study and parameters that can clarify the difference in the effect of the three biostimulants used, such as primary and secondary metabolites, should be studied.

Author Response

Dear Reviewer

as stated in the previous revision, this research was addressed to evaluate the potential agronomic benefits of three biostimulants (representing two of the most popular and widespread categories of non-microbial plant biostimulants: seaweed extracts and protein hydrolysates) on chicory root production in two agronomic trials under different controlled enviroonments. As far as we know this is the first agronomic study in the World dealing with biostimulants and chicory root production in controlled environment. For the above reasons, we belive that this study deserves to be published, and can represent a starting point for further studies on metabolomics as reported in the conclusions.

Best regards

Giuseppe Colla